# Risk prediction model for post-endoscopic retrograde cholangiopancreatography pancreatitis: A systematic review and meta-analysis

Yijun Mao[1], Qiang Liu[2]*, Hui Fan[1]*, Wenjing He[1]*, Cheng Zhang[3], Xueqian Ouyang[1], Erqing Li[4], Xiaojuan Wang[1], Li Qiu[1], Huanni Dong[1]

1 Department of Nursing, Xianyang Central Hospital, Xianyang, China, 2 Department of Orthopedic Surgery, Xianyang Central Hospital, Xianyang, China, 3 Department of Hepatobiliary Surgery, Xianyang Central Hospital, Xianyang, China, 4 Interventional Operating Room, Xianyang Central Hospital, Xianyang, China

* hlb33288602@163.com (QL); 240731187@qq.com (HF); 309772658@qq.com (WH)

## Abstract

### Background

Post-endoscopic retrograde cholangiopancreatography pancreatitis (PEP) is the most common and clinically significant complication of ERCP, with an incidence of 3.5–9.7% in general populations and up to 14.7% in high-risk groups, leading to considerable morbidity, mortality, and healthcare costs. Although numerous multivariable prediction models have been developed, their predictor sets, methodological rigor, and clinical applicability remain highly variable.

### Method

We conducted a PRISMA 2020–compliant systematic review and meta-analysis, prospectively registered in PROSPERO (CRD42024556967). Nine databases were searched to June 1, 2024, for studies developing or validating multivariable PEP risk prediction models. Data on study/model characteristics, predictors, and performance metrics were extracted. Risk of bias was assessed with PROBAST, and study quality with the Newcastle–Ottawa Scale. Random-effects meta-analyses pooled (i) PEP incidence, (ii) associations of individual predictors, and (iii) overall model performance.

### Results

Twenty-four studies (26 models; n = 38,016) published from 2002–2024 were included, predominantly retrospective cohorts from East Asia (n = 16). The pooled PEP incidence was 8.48% (95% CI: 6.90–10.39%; I² = 96.4%), highest in East Asia and retrospective cohorts. Strongest predictors included pancreatic duct cannulation

**Data availability statement:** All relevant data are within the manuscript and its Supporting Information files.

**Funding:** This research project was supported by Xianyang Science and Technology Planning Project (Grant Number: L2023-ZDYF-SF-055). The funders had no role in study design, data collection and analysis, decision to publish, or preparation of the manuscript.

**Competing interests:** The authors have declared that no competing interests exist.

(OR=3.50), pancreatic injection (OR=3.50), previous pancreatitis (OR=3.32), and pancreatic guidewire use (OR=2.63); additional consistent factors were female sex, difficult cannulation, elevated bilirubin, low albumin, choledocholithiasis, and prolonged procedure time. The pooled odds ratio for model performance was 0.81 (95% CI: 0.78–0.84; $I^2 = 83.5\%$), with AUCs ranging 0.560–0.915, though calibration was infrequently reported (38%) and external validation undertaken in only 46%. PROBAST indicated high overall risk of bias, chiefly in the analysis (92%) and participants (100%) domains.

## Conclusion

Current PEP prediction models generally demonstrate moderate-to-high discrimination but are limited by suboptimal calibration, inadequate external validation, and methodological heterogeneity. Future research should adhere to TRIPOD guidelines, employ multicenter large-sample designs, retain continuous predictors, address missing data with robust imputation methods, and conduct comprehensive temporal, geographic, and domain-specific validation. Integration of artificial intelligence/machine learning with conventional modeling and embedding validated tools into clinical workflows may enhance predictive accuracy and real-world utility.

## Introduction

Endoscopic retrograde cholangiopancreatography (ERCP) is widely utilized for diagnosing and treating biliopancreatic diseases [1,2]. Compared with traditional surgery, ERCP offers several advantages, including reduced surgical trauma, faster postoperative recovery, shorter treatment durations, and briefer hospital stays [3]. However, as an invasive diagnostic and therapeutic procedure, ERCP carries inherent risks, particularly procedure-related complications. Among these, post-ERCP pancreatitis (PEP) is the most common, with an incidence of 3.5–9.7% in the general population and up to 14.7% in high-risk patients. The mortality rate associated with PEP ranges from 0.1% to 0.7% [2,4,5]. While mild cases may simply prolong hospitalization, severe PEP can result in pancreatic necrosis, multiple organ failure, or even death [6,7]. These adverse outcomes not only compromise procedural success and patient prognosis but also impose substantial economic burdens on patients, healthcare systems, and society.

To mitigate these consequences, accurate identification of patients at high risk of PEP is essential. Risk prediction models offer clinicians a tool to estimate individual PEP risk and tailor preventive strategies accordingly. By stratifying patients according to predicted risk, unnecessary admissions can be reduced, preventive measures can be targeted, and overall healthcare costs can be lowered-ultimately improving patient quality of life.

In recent years, multiple PEP risk prediction models have been developed using different predictors, statistical methods, and validation strategies. However, their

methodological quality, external validity, and clinical applicability vary considerably. Moreover, the predictors incorporated into these models differ substantially across studies, reflecting variations in patient populations, procedural techniques, and study designs. While such heterogeneity is inevitable, it raises important questions: which predictors show consistent, clinically meaningful associations with PEP? How well do existing models perform overall? And what is the true incidence of PEP in the populations studied?

A systematic review and meta-analysis provides a rigorous approach to address these questions. By synthesizing available evidence, such an approach can: (i) quantify the pooled incidence of PEP across diverse populations; (ii) identify predictors with consistent, statistically significant associations with PEP; (iii) evaluate the overall predictive performance of existing models, considering both discrimination and calibration; (iv) assess the methodological quality and risk of bias of the included studies.

Therefore, the aim of this study was to systematically review and meta-analyze existing PEP risk prediction models, with a focus on their predictive performance, methodological rigor, and clinical applicability. Through this synthesis, we aim to provide robust, evidence-based recommendations to guide both clinical decision-making and future model development.

## Methods

This study was registered in PROSPERO (registration number: CRD42024556967) and conducted in accordance with the PRISMA 2020 guidelines. For detailed information, please refer to S1 Table.

### Search strategy

We systematically searched PubMed, Web of Science, The Cochrane Library, Embase, Cumulative Index to Nursing and Allied Health Literature (CINAHL), China National Knowledge Infrastructure (CNKI), Wanfang Database, China Science and Technology Journal Database (VIP), and SinoMed from inception to June 1, 2024. The detailed search strategies for each database are provided in S2 and S3 Tables.

### Inclusion and exclusion criteria

The inclusion criteria were as follows: (i) studies involving patients underwent ERCP; (ii) observational study design (cohort or case-control), or interventional studies that developed or validated a PEP prediction model; (iii) reported outcome of post-ERCP pancreatitis (PEP); and (iv) inclusion of a multivariable prediction model. The exclusion criteria were as follows: (i) studies that only assessed risk factors without constructing a prediction model; (ii) studies without available full text; (iii) gray literature, including conference abstracts and agency reports; (iv) duplicate publications; and (v) studies not written in English or Chinese.

### Study selection and screening

Two reviewers independently screened titles/abstracts and full texts according to the inclusion criteria. Disagreements were resolved by a third reviewer.

### Data extraction

Data were extracted into a standardized form and categorized as: (i) study characteristics: first author, year, country/region, research design, data source, study period, and PEP diagnostic criteria; (ii) model characteristics: sample size, outcome event rate, events per variable (EPV), model development method, variable selection method, handling of missing data, and treatment of continuous variables; (iii) model performance: discrimination (e.g., AUC/C-statistic), calibration, type of validation (internal vs. external), and model presentation format; and (iv) predictors: number and type of candidate variables and final predictors included in the model.

## Quality assessment

The methodological quality and applicability were assessed using the Prediction model Risk Of Bias ASsessment Tool (PROBAST) [8]. The Newcastle-Ottawa Scale (NOS) was employed to assess the quality of observational studies [9]. Two reviewers (YM. & QL.) independently rated the certainty of evidence using the GRADE system (Grading of Recommendations Assessment, Development and Evaluation) [10], with disagreements resolved through discussion.

## Data synthesis and statistic analysis

We performed three complementary meta-analyses:

1. Pooled incidence of PEP

Extracted incidence data (events/total) from each study. Proportions were pooled using a random-effects model with logit transformation to stabilize variances. Heterogeneity was assessed with Cochran's Q test and quantified by $I^2$; thresholds of 25%, 50%, and 75% indicated low, moderate, and high heterogeneity, respectively. Subgroup analyses were planned by geographic region, study design, diagnostic definition and validation method.

2. Pooled associations of individual predictors

For predictors reported in ≥3 studies, effect estimates (odds ratios [OR], relative risks [RR], hazard ratios [HR]) and 95% CIs were extracted; RRs and HRs were converted to ORs if necessary. Random-effects meta-analyses (restricted maximum likelihood estimator) were performed separately for each predictor. Heterogeneity and publication bias (Egger's test, funnel plots) were evaluated when ≥10 studies were available. Sensitivity analyses excluded high risk-of-bias studies or those with small sample sizes (<100 participants).

3. Pooled predictive performance of models

Discrimination was quantified by the AUC (C-statistic). AUC values were pooled on the logit scale and back-transformed for presentation. Priority was given to externally validated results; internal validation was used if external validation was unavailable.

All analyses were performed using R (version 4.4.1; R Foundation for Statistical Computing, Vienna, Austria) with the meta, metafor, and mada packages. Two-tailed p-values <0.05 were considered statistically significant.

## Results

### Study selection

Our search identified 962 records, with 57 duplicates removed. After screening 905 articles, 783 were excluded as irrelevant. An further 98 full-text articles were excluded for the following reasons: conference abstracts (n = 26), no risk prediction model (n = 36), fewer than two predictors (n = 2), abstract only (n = 23), and not primary literature (n = 11). In total, 24 studies [11–34] reporting 26 PEP prediction models met the inclusion criteria (Fig 1).

### Study characteristics

The 24 studies were published between 2002 and 2024, with 20 published in the last five years. Sixteen studies were conducted in China, four in Japan, two in the United States, one in Korea, and one in Italy. Study designs comprised 20 retrospective cohort, two prospective cohort, one cross-sectional study, and one case-control study. Diagnostic criteria for PEP varied: 13 studies used the Cotton Expert Consensus Criteria [35], nine the Atlanta Criteria [36], one the Chinese MDT Consensus [37], and one the ASGE criteria [38]. Sample sizes ranged from 312 to 6,731. The basic characteristics of the included studies are presented in Table 1.

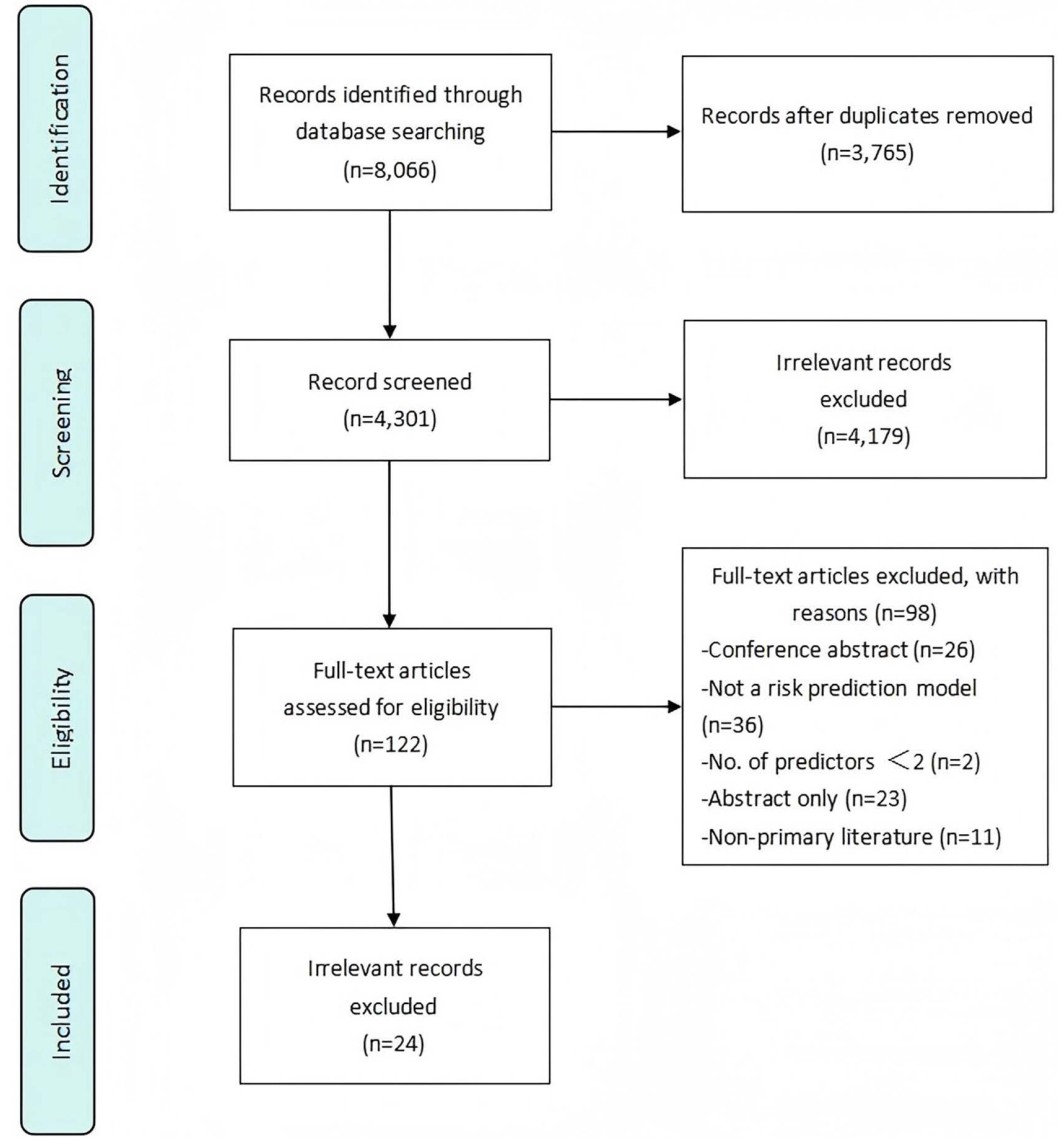

**Fig 1. Search results and study selection.**

Model development methods included logistic regression (n = 21), propensity score analysis (n = 1), gradient boosting (n = 1), and random forest (n = 1). Candidate predictors ranged from 3 to 22, the most frequent were pancreatic duct cannulation (n = 11), difficult cannulation (n = 10), pancreatic injection (n = 9), and female sex (n = 9) (Table 2 and Fig 2).

## Pooled incidence of PEP

Incidence data were available for 24 studies (n = 38,016 patients). The pooled incidence was 8.48% (95% CI: 6.90–10.39%) with high heterogeneity (I² = 96.4%) (Fig 3 and S1 Fig).

**By geographic region.** Subgroup analysis demonstrated significant geographical variation in PEP incidence (p = 0.0006). The highest incidence was observed in East Asia (9.27%, 95% CI: 7.40–11.55%; I² = 96.5%), followed by

## Table 1. Basic characteristics of the included studies.

| Author (year) | Country | Study design | Participants | Data source (Study period) | PEP diagnostic criteria | PEP cases/sample size (%) Training set | Validation set |
|---|---|---|---|---|---|---|---|
| Friedland 2002 [11] | US | PC | Patients undergoing ERCP | One hospital (1993–1998) | Cotton's criterion | 95/1705 (5.6%) | - |
| Dimagno 2013 [12] | US | CC | Patients undergoing ERCP | One teaching hospital (1997–2009) | Cotton's criterion | 211/5254 (4.0%) | - |
| Fang 2016 [13] | China | RC | Patients undergoing ERCP | The surgical ward of a teaching hospital (2014) | Cotton's criterion | 26/312 (8.3%) | - /108 |
| Wan 2018 [14] | China | RC | Patients undergoing ERCP | The medical ward of a teaching hospital (2012–2017) | Cotton's criterion | 44/740 (6.0%) | - |
| Chiba 2021 [15] | Japan | RC | Patients with suspected hepato-biliary-pancreatic disorders who underwent trans-papillary ERCP attempts | One teaching hospital (2012–2019) | Atlanta classification | 108/3362 (3.2%) | - |
| Dou 2021 [16] | China | RC | Patients undergoing ERCP | The surgical ward of a teaching hospital (2018–2020) | Cotton's criterion | 142/529 (26.8%) | 31/109 (28.4%) |
| Wang 2021 [17] | China | RC | Patients ≥18 years old undergoing ERCP | The surgical ward of a teaching hospital (2017–2019) | Cotton's criterion | 42/412 (10.2%) | - |
| Zhang 2021 | China | RC | Patients undergoing ERCP | The medical ward of a teaching hospital (2018–2021) | Atlanta classification | 61/1293 (4.7%) | - |
| Zheng 2021 [19] | China | RC | Patients ≥18 years old undergoing ERCP | One teaching hospital (2016–2019) | Atlanta classification | 104/1327 (7.8%)* 44/568 (7.8%)※ | 47/342 (13.7%) |
| Park 2021 [20] | South Korean | RC | Patients undergoing ERCP | Multicenter study (2015–2020) | Cotton's criterion | 42/760 (5.5%) | 29/735 (3.9%) |
| Fujita 2021 [21] | Japan | RC | Patients ≥18 years old undergoing ERCP | Multicenter study (2010–2012) | Cotton's criterion | 96/1969 (4.9%) | 37/750 (4.9%) |
| Zhang 2022 [18] | China | RC | Patients ≥18 years old undergoing ERCP for calculus of common bile duct with gallbladder in situ | The surgical ward of a teaching hospital (2010–2019) | Cotton's criterion | 128/986 (13.0%) | 57/540 (4.0%) |
| Fu 2022 | China | RC | Patients ≥18 years old undergoing ERCP for biliary stent placement due to MBO | One teaching hospital (2014–2021) | Cotton's criterion | 66/1016 (6.5%) | 42/508 (8.3%) |
| Huang 2022 [24] | China | RC | Patients > 18 years old undergoing ERCP | One teaching hospital (2019–2021) | Cotton's criterion | 66/372 (5.6%)* 25/186(7.5%)※ | 10/139 (7.2%) |
| Archibugi 2022 [54] | Italy | PC | Patients ≥18 years old undergoing ERCP | Multicenter study | Atlanta classification | 70/1150 (6.1%) | - |
| Ma 2023 [26] | China | RC | Patients > 18 years old undergoing ERCP | The medical ward of a hospital (2017–2022) | Atlanta classification | 45/603 (7.5%) | 23/205 (11.2%) |
| Yao 2023 [27] | China | RC | Patients > 18 years old undergoing ERCP | One hospital (2019–2021) | Atlanta classification | 41/404 (10.2%) | - |
| Qin 2023 [28] | China | RC | Patients ≥18 years old undergoing ERCP | The medical ward of a hospital (2017–2022) | Atlanta classification | 45/410 (11.0%) | 10/100 (10.0%) |
| Wang 2023 [29] | China | RC | Patients undergoing ERCP | The surgical ward of a hospital (2020–2022) | Chinese consensus | 105/1026 (10.2%) | - |

*(Continued)*

**Table 1.** (Continued)

| Author (year) | Country | Study design | Participants | Data source (Study period) | PEP diagnostic criteria | PEP cases/sample size (%) Training set | Validation set |
|---|---|---|---|---|---|---|---|
| Chen 2023 [30] | China | RC | Patients ≥18 years old undergoing ERCP with early postoperative hyperamylasemia | The surgical ward of a hospital (2018–2021) | Atlanta classification | 107/312 (37.3%) | - |
| Takahashi 2023 [31] | Japan | RC | Patients ≥20 years old undergoing ERCP | Multicenter study | Atlanta classification | 60/615 (10.2%) | 52/544 (9.6%) |
| Fukuda 2023 [32] | Japan | RC | Patients undergoing ERCP | Multicenter study (2010–2020) | ASGE's criterion | 159/2224 (7.2%) | 64/875 (7.3%) |
| Zhang 2023 [33] | China | RC | Patients undergoing ERCP | One teaching hospital (2010–2021) | Cotton's criterion | 241/4715 (5.1%) | - /2016 |
| Yan 2024 [34] | China | CS | Patients ≤60 years old undergoing ERCP with common bile duct stones | One hospital (2015–2023) | Cotton's criterion | 48/919 (5.2%) | - /81 |

Abbreviations: RC, Retrospective cohort; PC, Prospective cohort; CS, Cross sectional; CC, Case control; MBO, malignant biliary obstruction; ASGE, American Society for Gastrointestinal Endoscopy; "-", not reported.

* indicate training set; ※ indicate test set.

Europe (6.09%, 95% CI: 4.84–7.62%; single study), North America (4.69%, 95% CI: 3.40–6.45%; $I^2$=86.4%), and other regions (4.74%, 95% CI: 3.40–6.57%; $I^2$=51.1%) (S2 Fig).

**By study design.** When stratified by study design, retrospective cohort studies reported the highest incidence (9.05%, 95% CI: 7.24–11.25%; $I^2$=96.4%), followed by prospective cohort studies (5.78%, 95% CI: 4.98–6.70%; $I^2$=0%), cross-sectional studies (5.22%, 95% CI: 3.96–6.86%; single study), and case–control studies (4.02%, 95% CI: 3.52–4.58%; single study). The difference between subgroups was statistically significant (p<0.0001) (S3 Fig).

**By diagnostic definition.** The pooled incidence was 9.26% (95% CI: 6.25–13.51%; $I^2$=96.8%) in studies using the Atlanta definition, 10.23% (95% CI: 8.52–12.24%; single study) in those adopting the Chinese definition, and 7.96% (95% CI: 6.14–10.27%; $I^2$=96.6%) in studies applying other definitions. No statistically significant difference was observed between subgroups (p=0.2919) (S4 Fig).

**By validation method.** Marked variation was also detected when stratified by validation approach (p<0.0001). Studies with external validation only reported the highest incidence (27.12%, 95% CI: 23.81–30.70%; $I^2$=0%), followed by those with both internal and external validation (10.63%, 95% CI: 7.96–14.08%; $I^2$=83.8%), no validation (8.55%, 95% CI: 4.92–14.46%; $I^2$=98.1%), and internal validation only (6.75%, 95% CI: 5.61–8.10%; $I^2$=91.8%) (S5 Fig).

## Meta-analysis of individual predictors

The pooled effects of 11 risk factors for PEP are presented in Table 3 and S6 Fig, including odds ratios (ORs), 95% confidence intervals (CIs), and heterogeneity metrics ($I^2$ and Tau$^2$). Subgroup analyses by region (East Asia vs. North America) were performed where data permitted.

**High-risk factors with moderate heterogeneity.** Pancreatic duct cannulation (OR = 3.50, 95% CI: 1.93–6.34) and pancreatic injection (OR = 3.50, 95% CI: 1.71–7.14) showed the strongest associations, though with substantial heterogeneity ($I^2$=88% for both). Difficult cannulation had a higher effect size in North America (OR = 3.44, 95% CI: 1.34–8.81, $I^2$=0%) compared to East Asia (OR = 2.56, 95% CI: 1.54–4.27, $I^2$=77%).

**Factors with low heterogeneity.** Previous pancreatitis (OR = 3.32, 95% CI: 2.37–4.64, $I^2$=25%) and PGW (OR = 2.63, 95% CI: 2.02–3.43, $I^2$=6%) were statistically significant with minimal heterogeneity.

**Table 2. Domains of predictors and performance of stroke readmission risk prediction models.**

| Author (year) | Missing data handling | Continuous variable processing method | Model development method | Calibration method | Validation method | Final predictors | Model performance | Model presentation |
|---|---|---|---|---|---|---|---|---|
| Friedland 2002 [11] | Delete | Categorical variables | LR | - | - | Pain during the procedure, cannulation of the pancreatic duct, previous post-ERCP pancreatitis, number of cannulation attempts | - | Point score system |
| Dimagno 2013 [12] | - | Continuous variables | LR | - | - | Current smoking, CLD-biliary, CLD-transplant/hepatectomy complications, younger age, suspected SOD, pancreatic sphincterotomy, MDC | A: 0.740 | Point score system |
| Fang 2016 [13] | - | Categorical variables | LR | - | Internal validation & temporal validation | Prothrombin time, type of duodenal papilla, repeatedly entering the duodenal papilla(> 2) | A: 0.873 (0.795,0.952) B: 0.862 (0.760,0.964) | Formula of risk score obtained by regression coefficient of each factor |
| Wan 2018 [14] | - | Categorical variables | LR | - | - | Female, history of hypertension, first time to ERCP, not completely taking out calculuses, difficult intubation, pancreatograpgy | A: 0.706 (0.677,0.735) | Point score system |
| Chiba 2021 [15] | - | Categorical variables | LR, PS | - | Internal validation | Naïve papilla, PGW, difficult cannulation (> 15 min), pancreatic injections (≥1), absence of pancreatic stent | A: 0.86 (0.82,0.89) B: 0.81 (0.77,0.86) | Point score system |
| Dou 2021 [16] | - | Continuous variables | LR | Hosmer-Lemeshow test | Geographical validation | Female, age, previous pancreatitis, previous cholecystectomy, operation time, multiple pancreatic duct intubation | A: 0.869 | Nomogram model |
| Wang 2021 [17] | - | Categorical variables | LR | - | - | Female, pancreatic tube development, difficult cannulation, operating time ≥ 45 min | A: 0.856 | point score system |
| Zhang 2021 | Delete | Continuous variables | LR | Hosmer-Lemeshow test | - | Small papilla, diverticula at the root of papilla, misdirection of guide wire into pancreatic duct, cholecystolithiasis, cholangitis, metallic stent placement in bile duct, normal upper limit of TBIL | A: 0.834 (0.783,0.886) | Nomogram model |
| Zheng 2021 [19] | - | Categorical variables | LR | - | Internal validation & temporal validation | Gastrectomy history, high DBIL, high ALB, villus type of papillary orifice, nodular type of papillary orifice, pancreatic guidewire passages, precut, expert operators | A: 0.793 B: 0.718 | Formula of risk score obtained by regression coefficient of each factor |
| Park 2021 [20] | Delete | Continuous variables | LR | - | Internal validation & domain validation | Age ≤ 65, female, acute pancreatitis history, malignant biliary obstruction, pancreatic sphincterotomy | - | Point score system |
| Fujita 2021 [21] | - | Categorical variables | LR | Hosmer-Lemeshow test | Internal validation & domain validation | History of post-ERCP pancreatitis, intact papilla, difficult cannulation, PGW-assisted biliary cannulation, pancreatic injection, pancreatic IDUS/sampling from the pancreatic duct, biliary IDUS/sampling from the biliary duct | A: 0.799 B: 0.791 | Point score system |

*(Continued)*

**Table 2.** (Continued)

| Author (year) | Missing data handling | Continuous variable processing method | Model development method | Calibration method | Validation method | Final predictors | Model performance | Model presentation |
|---|---|---|---|---|---|---|---|---|
| Zhang 2022 [18] | - | Categorical variables | LR | - | Internal validation & domain validation | History of pancreatitis, increment of white blood cell count, guide wire into the pancreatic duct, mechanical 1ithotripsy, gallbladder stone, gallbladder wall thickening | A: 0.691 (0.642,0.740) B: 0.747 (0.687,0.806) | Nomogram model |
| Fu 2022 | - | Categorical variables | LR | Hosmer-Leme-show test | Internal validation | Acute pancreatitis history, the absence of pancreatic duct dilation, nonpancreatic cancer, difficult cannulation, pancreatic injection | A: 0.810 (0.751,0.868) B: 0.781 (0.703,0.858) | Point score system |
| Huang 2022 [24] | - | Continuous variables | LR | Calibration curve | Internal validation & temporal validation | BMI, bile duct stricture, biliary sphincter balloon dilatation, guide wire inserted into pancreatic duct, development of pancreatic duct | A: 0.848 B: 0.933 | Nomogram model |
| Archibugi 2022 [54] | - | Categorical variables | GB, LR | - | Internal validation | Bilirubin, age, BMI, procedure time, previous sphincterotomy, alcohol units/day, cannulation attempts, female, gallstones, use of Ringer's solution, periprocedural NSAIDs | A: 0.7±0.076 (0.64,0.76) B: 0.671 | - |
| Ma 2023 [26] | - | Continuous variables | LR | - | Internal validation & domain validation | History of gastrectomy, calculus of common bile duct, papillary foramen nodule type, pancreatic wire channel, sphincterotomy, elevated TBIL, decreased ALB | A: 0.895 (0.811,0.934)] B: 0.864 (0.802,0.911) | Point score system |
| Yao 2023 [27] | - | Categorical variables | LR | - | - | Pancreatitis history, pancreatic duct visualization, EST, intubation difficulty, putting guide wire into the pancreatic duct frequently | A: 0.804 (0.762,0.841) | Formula of risk score obtained by regression coefficient of each factor |
| Qin 2023 [28] | - | Continuous variables | LR | - | Internal validation & temporal validation | Choledocholithiasis, nodule type of papillary orifice, pancreatic guidewire channel, TBIL over or equal to 250 μmol/L, ALB less than 35g/L | A: 0.895 (0.812,0.944) B: 0.864 (0.801,0.936) | Nomogram model |
| Wang 2023 [29] | - | Categorical variables | LR | Hosmer-Leme-show test | - | Pancreatic duct intubation, frequency of contrast medium injection ≥2 times, history of pancreatitis, and sphincter dysfunction | A: 0.842 (0.807,0.877) | Formula of risk score obtained by regression coefficient of each factor |
| Chen 2023 [30] | - | Categorical variables | LR | Hosmer-Leme-show test | - | Female, age below 60 years, stenosis of inferior common bile duct, difficult biliary cannulation, absence of pancreatic duct stent, and surgical time over or equal to 1 hour | A: 0.812 (0.760,0.865) | Nomogram model |
| Takahashi 2023 [31] | - | Categorical variables | RF, LR | - | Internal validation & domain validation | Albumin, creatinine, biliary tract cancer, pancreatic cancer, bile duct stone, total procedure time, pancreatic duct injection, PGW without a pancreatic stent, IDUS, bile duct biopsy | A: 0.821 B: 0.770 | Point score system |

*(Continued)*

**Table 2.** (Continued)

| Author (year) | Missing data handling | Continuous variable processing method | Model development method | Calibration method | Validation method | Final predictors | Model performance | Model presentation |
|---|---|---|---|---|---|---|---|---|
| Fukuda 2023 [32] | - | Continuous variables | LR | Calibration curve | Internal validation & domain validation | Female, indication for ERCP, difficult cannulation, guidewire insertion into the pancreatic duct, Endoscopic sphincterotomy or sphincteroplasty | A: 0.720 | Nomogram model |
| Zhang 2023 [33] | - | Categorical variables | LR | Calibration curve | Internal validation | Female, ERCP indication, cannulation method, cannulation time, number of cannulation attempts, unintended pancreatic duct cannulation, and no trainees involvement in the cannulation | A: 0.700 | Nomogram model |
| Yan 2024 [34] | - | Continuous variables | LR, Lasso | - | Internal validation & domain validation | Female, HBP, DM, history of pancreatitis, history of hepatitis, presence of the gallbladder, with gallstone, number of stones, type of duodenal papillae, difficult cannulation, EST, EPBD, age (33–50 years), hemoglobin > 131 g/L, platelet < 203.04 or > 241.40 × 109/L, neutrophil percentage > 58.90%, TBIL > 18.39 umol/ L, aspartate amino transferase < 36.56 IU/ L, alkaline phosphatase < 124.92 IU/ L, albumin < 42.21 g/ L, common bile duct diameter (7.25–10.02 mm), diameter of stone | A: 0.826 (0.772,0.881) B: 0.838 (0.689,0.986) | Website online calculator |

Abbreviations: LR, Logistic regression; PS, Propensity Score analysis; GB, Gradient Boosting; RF, random forest; Lasso, Lasso regression; CLD, chronic liver disease; SOD, suspected sphincter of Oddi dysfunction; MDC, moderate-difficult cannulation; PGW, pancreatic guidewire assisted technique; IDUS, intraductal ultrasonography; BMI, Body mass index; NSAIDs, non steroidalanti inflammatorydrugs; HBP, high blood pressure; DM, diabetes mellitus; EST, Endoscopic Sphincterotomy; EPBD, endoscopic papillary balloon dilatation; TBIL, total bilirubin; DBIL, direct bilirubin; ALB, albumin.

"-", not reported; A, development cohort; B, validation cohort.

**Non-significant factors.** Age, TBIL, calculus of common bile duct, and operation time had wide CIs crossing 1, despite high heterogeneity ($I^2 > 70\%$).

**Heterogeneity and bias assessment.** High heterogeneity ($I^2 > 75\%$) was observed for pancreatic duct cannulation, pancreatic injection, and female gender, possibly due to procedural differences across studies. Publication bias (Egger's test) was significant only for pancreatic duct cannulation (p = 0.039), warranting caution in interpretation.

## Predictive performance of models

Our systematic review and meta-analysis included 13 studies evaluating risk prediction models for ERCP pancreatitis. The pooled analysis demonstrated that these prediction models had significant predictive value, with an overall odds ratio (OR) of 0.80 (95% CI: 0.77–0.84) using a random-effects model. Considerable heterogeneity was observed among studies ($I^2 = 82.0\%$, $\tau^2 = 0.0041$, p < 0.0001), suggesting substantial variation in model performance across different settings (Fig 4).

## Subgroup analyses

**Dataset type.** Analysis stratified by dataset type (development vs. validation groups) revealed no significant difference between subgroups ($\chi^2_1 = 1.44$, p = 0.2303). The development group showed an OR of 0.78 (95% CI: 0.72–0.85, $I^2 = 89.8\%$), while the validation group demonstrated an OR of 0.82 (95% CI: 0.78–0.87, $I^2 = 47.3\%$) (S7A Fig).

**Diagnostic criteria.** Subgroup analysis by diagnostic criteria (Atlanta classification, Chinese consensus, and Cotton criterion) showed no statistically significant differences between subgroups ($\chi^2_2 = 4.57$, p = 0.1020). Models using Atlanta

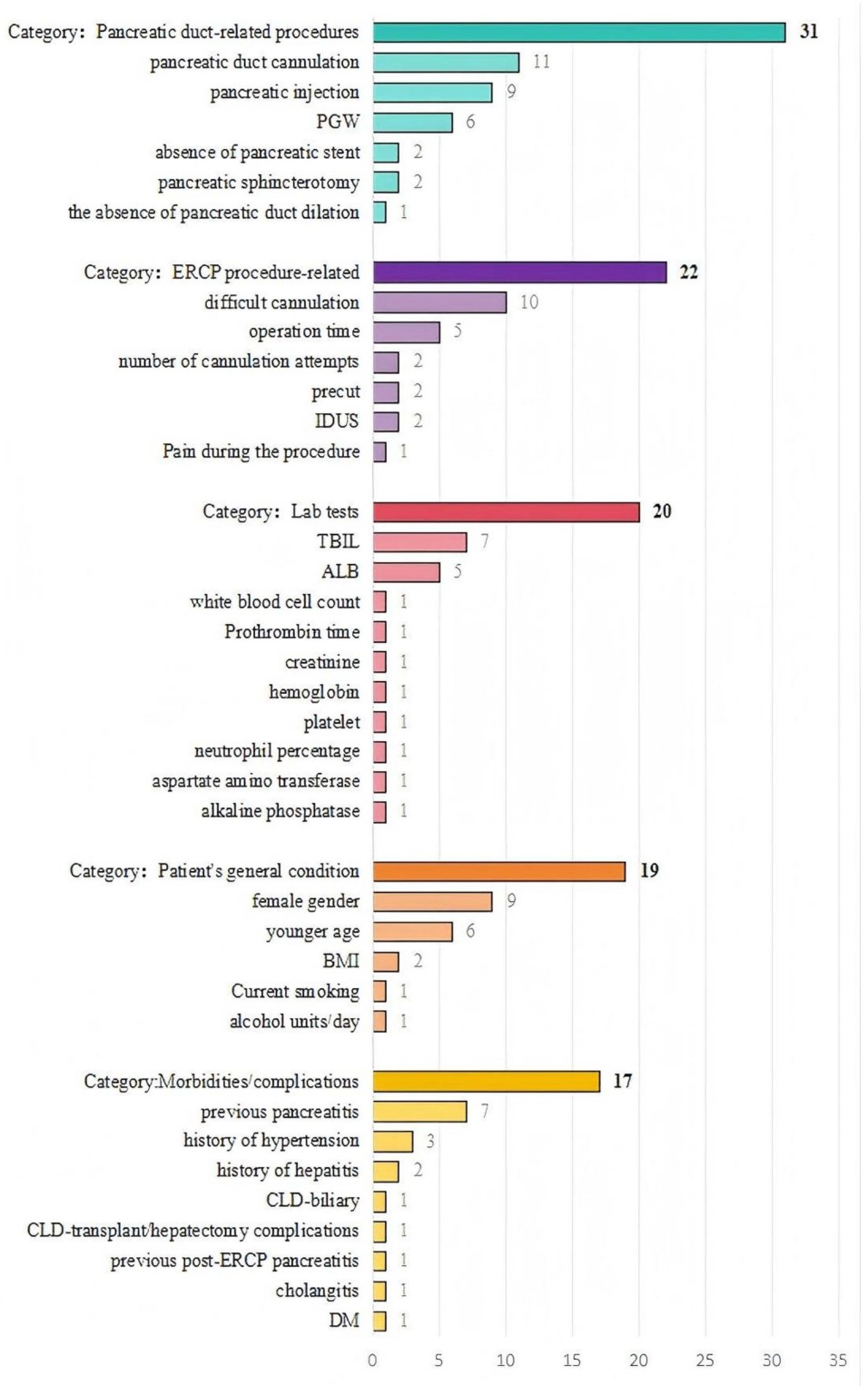

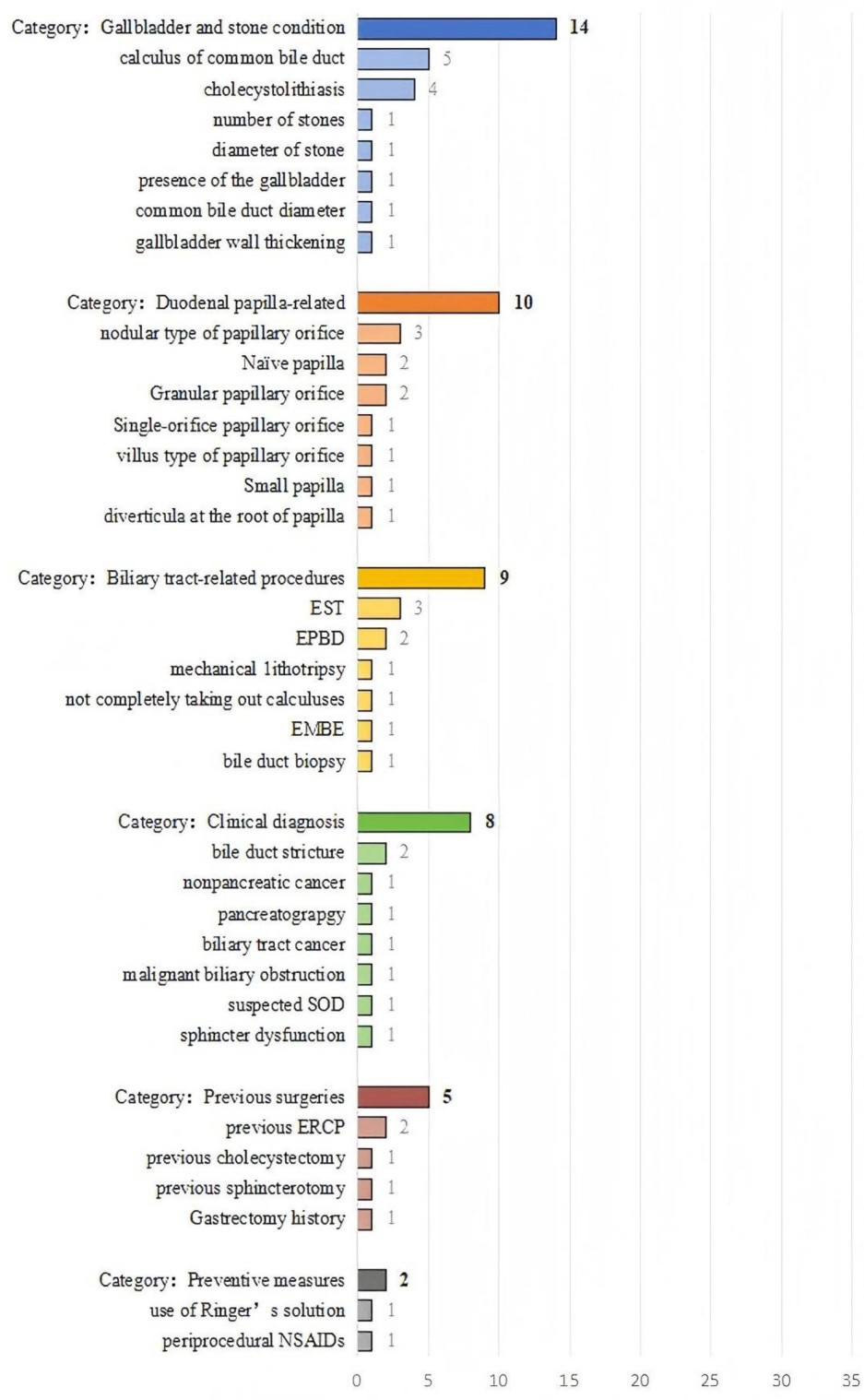

**Fig 2. Predictors included in the final development models.**

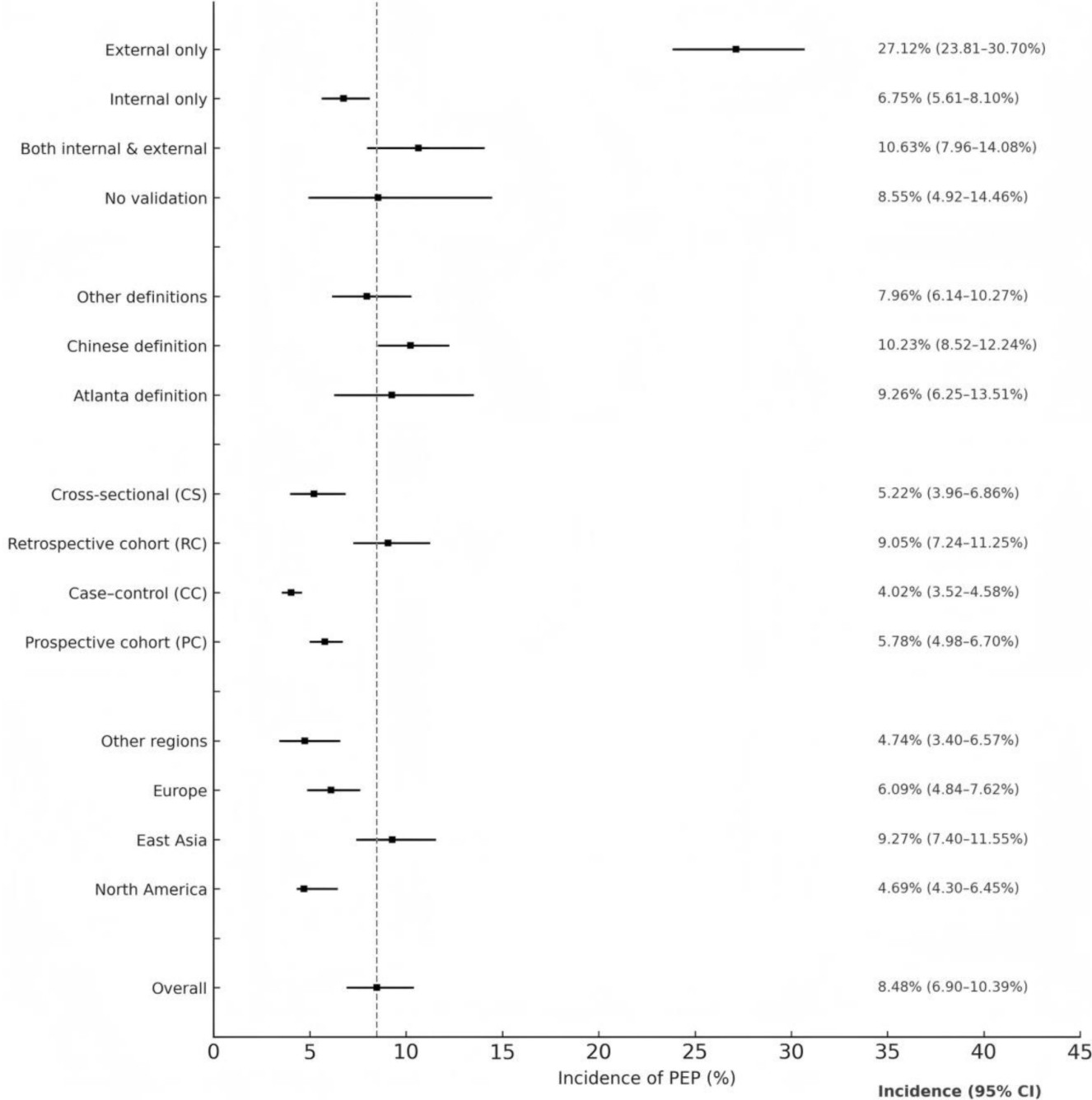

**Fig 3. Pooled incidence of post-endoscopic retrograde cholangiopancreatography pancreatitis (PEP) by predefined subgroups.** This forest plot presents pooled incidence estimates (%) and 95% confidence intervals (CIs) derived from random-effects meta-analyses using the inverse variance method and logit transformation. Subgroups are arranged into four thematic categories: region, study design, outcome definition, and validation method. The vertical dashed line represents the overall pooled incidence across all included studies (n = 37; 38,016 patients), estimated at 8.48% (95% CI: 6.90–10.39%). Point estimates are indicated by circles, with horizontal lines showing the 95% CIs. Numerical values to the right of each line represent the pooled proportion and its 95% CI. Heterogeneity within subgroups was quantified using the I² statistic, and between-group differences were evaluated using the Q test. The highest incidence was observed in East Asia and in studies with external validation only, whereas the lowest incidence occurred in North America and case–control study designs.

**Table 3. Key findings from the meta-analysis of PEP risk factors.**

| Factors | No studies | OR (95%CI) | I² (%) | Tau² | Comparison | Egger's p | Subgroup (Region) |
|---|---|---|---|---|---|---|---|
| Pancreatic duct cannulation | 10 | 3.50 (1.93-6.34) | 88 | 0.572 | Yes vs No | 0.039 | East Asia: OR = 3.50 (1.93–6.34) |
| Pancreatic injection | 7 | 3.50 (1.71-7.14) | 88 | 0.516 | Yes vs No | NA | East Asia: OR = 4.07 (1.86–8.92) |
| Previous pancreatitis | 7 | 3.32 (2.37-4.64) | 25 | 0.061 | Yes vs No | NA | East Asia: OR = 3.32 (2.37–4.64) |
| Difficult cannulation | 9 | 2.73 (1.87-3.99) | 75 | 0.176 | Yes vs No | NA | East Asia: OR = 2.56 (1.54–4.27) North America: OR = 3.44 (1.34–8.81) |
| PGW | 6 | 2.63 (2.02-3.43) | 6 | 0.014 | Yes vs No | NA | East Asia: OR = 2.63 (2.02–3.43) |
| Female gender | 7 | 2.25 (1.33-3.80) | 91 | 0.264 | Female vs Male | NA | East Asia: OR = 2.58 (1.56–4.27) |
| Age | 4 | 1.63 (0.58-4.63) | 89 | 0.376 | Young vs Old | NA | East Asia: OR = 1.98 (0.35–11.18) |
| Operation time | 3 | 1.45 (0.46-4.56) | 71 | 0.163 | Long vs Short | NA | East Asia: OR = 1.45 (0.46–4.56) |
| ALB | 2 | 1.26 (0.84-1.89) | 0 | 0 | Low vs High | NA | East Asia: OR = 1.26 (0.84–1.89) |
| Calculus of common bile duct | 4 | 1.19 (0.43-3.27) | 85 | 0.345 | Yes vs No | NA | East Asia: OR = 1.19 (0.43–3.27) |
| TBIL | 5 | 1.13 (0.58-2.19) | 90 | 0.242 | Low vs High | NA | East Asia: OR = 1.13 (0.58–2.19) |

Abbreviations: ALB, albumin; PGW, pancreatic guidewire assisted technique; TBIL, total bilirubin.

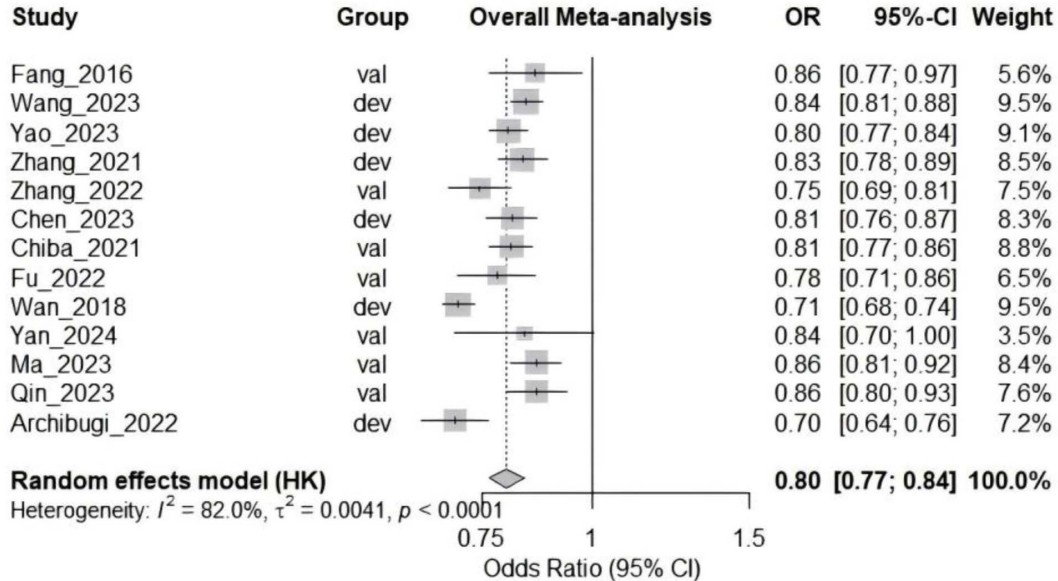

**Fig 4. Forest plot of the meta-analysis of pooled AUC estimates for predictive models performance.**

classification had an OR of 0.81 (95% CI: 0.76–0.86, $I^2 = 67.2\%$), while those using Cotton criterion showed an OR of 0.77 (95% CI: 0.70–0.85, $I^2 = 71.9\%$) (S7B Fig).

**Study design.** Significant differences were observed across study designs ($\chi^2_3 = 45.21$, p < 0.0001). Retrospective cohort studies (OR = 0.82, 95% CI: 0.80–0.85, $I^2 = 34.3\%$) constituted the majority of included studies. Cross-sectional studies showed higher predictive values (OR = 0.84, 95% CI: 0.70–1.00, single study), while prospective cohort studies demonstrated lower values (OR = 0.70, 95% CI: 0.64–0.76) (S7C Fig).

**Validation method.** Models with both internal and external validation showed the most consistent results (OR = 0.86, 95% CI: 0.85–0.88, $I^2 = 0\%$), differing significantly from those with internal validation only (OR = 0.79, 95% CI: 0.72–0.85, $I^2 = 72.5\%$) or no validation (OR = 0.80, 95% CI: 0.73–0.87, $I^2 = 90.5\%$) ($\chi^2_2 = 14.64$, p = 0.0007) (S7D Fig).

**Geographic region.** Significant regional variations were observed ($\chi^2_2 = 46.69$, p < 0.0001). East Asian studies predominated (OR = 0.82, 95% CI: 0.80–0.85, $I^2 = 27.2\%$), while European (OR = 0.70, 95% CI: 0.64–0.76) and North American studies (OR = 0.71, 95% CI: 0.68–0.74) showed lower predictive values (S7E Fig).

**Publication year.** No significant difference was found between studies published in 2016−2020 (OR = 0.77, 95% CI: 0.22–2.74, $I^2 = 89.7\%$) and those published in 2021−2024 (OR = 0.81, 95% CI: 0.78–0.84, $I^2 = 61.6\%$) ($\chi^2_1 = 0.20$, p = 0.656) (S7F Fig).

**Publication bias and sensitivity analysis.** Visual inspection of the funnel plot showed symmetrical distribution of studies (S7G Fig), and Egger's test confirmed no significant publication bias (t = 0.54, df = 11, p = 0.5967; bias estimate = 1.0742, SE = 1.7913). Sensitivity analysis using the leave-one-out method demonstrated robust results, with the pooled OR ranging from 0.79 (95% CI: 0.76–0.82) to 0.80 (95% CI: 0.77–0.84) when individual studies were sequentially removed, indicating that no single study disproportionately influenced the overall results (S7H Fig).

Calibration was assessed in nine studies: six reported good calibration (Hosmer–Lemeshow p > 0.05), and three used calibration curves, two of which showed moderate deviation (slope < 1.0, intercept > 0). One high-discrimination model exhibited poor calibration, suggesting overfitting.

## Model validation

Fifteen studies underwent internal validation (random split: n = 7; cross-validation: n = 4; bootstrapping: n = 2), and 11 included both internal and external validation.

## Risk of bias and applicability

The "Participants" domain was assessed as having a high risk of bias in all studies, primarily because retrospective designs may introduce information bias due to the unsystematic collection of predictor and outcome data, which is not ideal for prognostic modeling. Prospective cohort studies, which follow a longitudinal temporal relationship between predictors and outcomes, are considered the optimal study design, as they capture disease progression in its natural state.

For the "Predictors" domain, two studies were assessed as having a low risk of bias, particularly those utilizing prospective cohort designs, where predictors were measured before outcomes occurred. However, 22 studies were assessed as having an unclear risk of bias because they did not specify whether predictors were evaluated independently of outcome knowledge.

In the "Outcome" domain, two studies were assessed as having a low risk of bias due to the retrospective design, where outcomes were determined before predictor measurements, potentially linking outcome determination to the predictor. The remaining 22 studies were assessed as having an unclear risk of bias due to insufficient information on whether predictor data was available at the time of outcome determination.

In the "Analysis" domain, 22 studies were assessed as having a high risk of bias. Two studies had insufficient sample sizes, failing to meet the criterion that the number of clinical outcome events should be 20 times greater than the number of potential predictors. Fifteen studies converted continuous variables into categorical ones, leading to potential information loss, abrupt changes in predictions near thresholds, reduced statistical efficacy, and decreased result credibility. Three studies directly deleted missing data, which may have introduced selection bias and resulted in information loss. Additionally, 21 studies filtered predictors based on univariate analysis, potentially overemphasizing statistical significance while neglecting non-significant variables that might still contribute to prediction.

Regarding applicability, in the "Patients" domain, two studies included patients with choledocholithiasis undergoing ERCP, one study included patients with malignant biliary obstruction undergoing ERCP, and one study included patients with early onset hyperamylasemia after ERCP treatment. These studies had a poor risk of applicability to the broader

study population. The remaining 20 studies were assessed as having a low risk of applicability. The "Predictors" and "Outcome" domains were assessed as having low concerns about applicability in all studies (Table 4).

While PROBAST remains our primary tool given its specificity for prediction models, supplementary assessment with NOS revealed that all 24 studies met high-quality criteria (NOS ≥ 7). The study-specific NOS scores are available in S4 Table.

## Certainty of evidence (GRADE)

GRADE assessment indicated an overall moderate certainty of evidence. Evidence profiles generated by GRADEpro software are available in S5 Table.

**Risk of bias.** The included prediction models adhered to rigorous standards in predictor measurement and outcome ascertainment. However, some studies failed to report model calibration details, leading to an overall moderate risk of bias (downgraded by one level).

**Indirectness.** While most models were derived from tertiary center cohorts, limiting generalizability to primary care settings, the universal applicability of predictor variables (e.g., procedural difficulty scores) justified no downgrade.

**Inconsistency.** Substantial heterogeneity in model discrimination was observed ($I^2 = 82\%$), likely due to variations in endoscopist expertise (downgraded by one level).

**Table 4. Risk of bias appraisal results of eligible articles adapted from PROBAST.**

| Study | ROB | | | | Applicability | | | Overall | |
|---|---|---|---|---|---|---|---|---|---|
| | Participants | Predictors | Outcome | Analysis | Participants | Predictors | Outcome | ROB | Applicability |
| Friedland 2002 [11] | + | + | + | - | + | - | + | - | - |
| Dimagno 2013 [12] | - | ? | ? | - | - | + | + | - | + |
| Fang 2016 [13] | - | ? | ? | - | + | + | + | - | + |
| Wan 2018 [14] | - | ? | ? | - | + | + | + | - | + |
| Chiba 2021 [15] | - | ? | ? | - | + | + | + | - | + |
| Dou 2021 [16] | - | ? | ? | - | + | + | + | - | + |
| Wang 2021 [17] | - | ? | ? | - | + | + | + | - | + |
| Zhang 2021 | - | ? | ? | - | + | + | + | - | + |
| Zheng 2021 [19] | - | ? | ? | - | + | + | + | - | + |
| Park 2021 [20] | - | ? | ? | - | + | + | + | - | + |
| Fujita 2021 [21] | - | ? | ? | - | + | + | + | - | + |
| Zhang 2022 [18] | - | ? | ? | - | + | + | + | - | + |
| Fu 2022 | - | ? | ? | - | + | + | + | - | + |
| Huang 2022 [24] | - | ? | ? | + | + | + | + | - | + |
| Archibugi 2022 [54] | + | + | + | - | + | + | + | - | + |
| Ma 2023 [26] | - | ? | ? | - | + | + | + | - | + |
| Yao 2023 [27] | - | ? | ? | - | + | + | + | - | + |
| Qin 2023 [28] | - | ? | ? | - | + | + | + | - | + |
| Wang 2023 [29] | - | ? | ? | - | + | + | + | - | + |
| Chen 2023 [30] | - | ? | ? | - | + | + | + | - | + |
| Takahashi 2023 [31] | - | ? | ? | - | + | + | + | - | + |
| Fukuda 2023 [32] | - | ? | ? | - | + | + | + | - | + |
| Zhang 2023 [33] | - | ? | ? | - | + | + | + | - | + |
| Yan 2024 [34] | - | ? | ? | ? | - | + | + | - | - |

Abbreviations: " - ", High risk of bias; " + ", Low risk of bias; "?", unclear.

**Imprecision.** The pooled C-statistic (0.80, 95% CI 0.78–0.84) did not exceed the prespecified clinical decision threshold (0.75), thus no downgrade was warranted.

**Publication bias.** Funnel plot symmetry (Egger's P = 0.5967) indicated no significant publication bias.

## Discussion

### Principal findings

This systematic review and meta-analysis synthesized evidence from 24 studies on post-ERCP pancreatitis (PEP) prediction models. Three key findings emerged: (i) the pooled incidence of PEP highlights its substantial clinical burden, particularly among high-risk patient groups; (ii) several predictors—including pancreatic duct cannulation, difficult cannulation, pancreatography, female sex, pancreatic duct guidewire use, history of pancreatitis, total bilirubin, albumin, and choledocholithiasis—show consistent and significant associations with PEP risk; (iii) while many models achieved good discrimination (AUC > 0.80), calibration reporting and external validation were often insufficient, and overall methodological quality varied considerably.

### PEP incidence and clinical implications

Our pooled estimates confirm that PEP remains a frequent complication of ERCP, with higher incidence in high-risk groups. This reinforces the need for risk-stratified prevention strategies and justifies the use of predictive models to identify patients who may benefit most from prophylactic interventions. Although incidence estimates varied, much of the heterogeneity can be explained by procedural factors, operator experience, and patient selection criteria—highlighting the importance of adjusting models for these variables.

### Predictors and underlying mechanisms

Across the included studies, the most frequent predictors were pancreatic duct cannulation, difficult cannulation, pancreatography, female sex, history of pancreatitis, elevated total bilirubin, low albumin, pancreatic duct guidewire use, longer procedure time, and choledocholithiasis.

Mechanistically, these factors may cause mechanical or chemical injury to the pancreatic duct, induce papillary congestion or Oddi sphincter spasm, increase intraductal pressure, or promote premature activation of pancreatic enzymes. For example, repeated or difficult cannulation can exacerbate mucosal edema and obstruct pancreatic fluid outflow, while contrast injection during pancreatography may create high intraductal pressure, leading to reflux of pancreatic juice into the parenchyma and triggering autodigestion [39,40]. Female patients may be more susceptible due to a higher prevalence of sphincter of Oddi dysfunction [41]. A history of pancreatitis may reflect latent injury to the pancreaticobiliary system, increasing vulnerability to procedure-related trauma. Low albumin levels may indicate impaired protein metabolism and reduced tissue resilience, whereas elevated total bilirubin may reflect pancreaticobiliary dysfunction; both have been variably linked to PEP risk [42–44], though inconsistent findings suggest the need for further validation in large, multicenter studies.

Notably, our analysis and prior systematic reviews consistently identify pancreatic duct cannulation and female sex as robust predictors [45,46], with the latter possibly linked to sex-specific differences in sphincter of Oddi tone and reactivity [47–49]. Beyond individual predictors, our findings highlight the importance of synergistic effects: combined mechanical insults from cannulation, pancreatography, and guidewire use may produce cumulative injury, substantially elevating PEP risk compared to any single factor alone [50–52]. Prolonged procedure time may further amplify these effects, especially in patients with compromised metabolic or structural resilience [53].

Given these observations, future predictive models should explicitly incorporate such interactions, either through interaction terms in traditional regression or via non-linear modeling approaches, to better capture the complex interplay of patient-, procedure-, and operator-related factors and improve risk stratification accuracy [54,55].

## Predictive model performance and methodological quality

Across the included studies, model discrimination exhibited wide variability (AUC range: 0.560–0.915), with 58.3% (14/24) achieving an AUC greater than 0.80. Higher-performing models generally incorporated multi-domain predictors—encompassing biochemical, procedural, and anatomical variables—whereas lower-performing models relied predominantly on demographic characteristics and basic laboratory parameters. Nevertheless, discrimination alone does not ensure accurate absolute risk estimation. Calibration, which evaluates the concordance between predicted and observed probabilities, was reported in only nine studies, and calibration plots—currently the most informative method—were seldom employed. Several models with high AUC values demonstrated poor calibration, indicating potential overfitting and limited clinical transportability.

Our PROBAST assessment identified several recurring methodological limitations: (i) Many models failed to satisfy the widely recommended threshold of at least 20 events per variable (EPV), which is critical to reduce overfitting and enhance generalizability. For instance, a model with 10 predictors should ideally be based on ≥200 outcome events to ensure statistical stability; (ii) Continuous predictors were frequently dichotomized or converted into categorical variables, leading to loss of information and reduced predictive power. More robust strategies include retaining continuous variables in their original form, applying restricted cubic splines, or determining optimal cutoffs using statistical indices such as the Youden index from ROC analysis; (iii) Complete-case analysis was frequently adopted, which risks introducing bias and diminishing statistical efficiency. Multiple imputation, accompanied by sensitivity analyses, is preferable to maximize data utilization and maintain the validity of results; (iv) The common practice of univariate pre-screening can exclude clinically relevant variables, as statistical insignificance in univariate analysis does not preclude predictive value in multivariable contexts. Candidate predictors should be selected based on prior literature, established pathophysiological mechanisms, and clinical reasoning, alongside statistical considerations; (v) Sixteen studies did not perform internal validation; bootstrap resampling is recommended, particularly for small datasets, as it avoids the sample reduction inherent to cross-validation. External validation was conducted in only 12 studies, limiting generalizability to broader populations and diverse clinical settings.

When integrated with our meta-analytic findings, the pooled estimates of model performance, the overall incidence of PEP, and the effect sizes of individual predictors collectively reveal marked heterogeneity in model design and quality. These observations highlight the necessity for standardized methodological frameworks, rigorous internal and external validation, and transparent reporting to facilitate the development of robust, clinically applicable prediction models for PEP.

## Methodological implications

These findings directly support the appropriateness of our systematic review and meta-analysis approach. Although predictor heterogeneity is inevitable, pooling allowed us to (i) identify stable, high-impact predictors; (ii) quantify model performance; and (iii) estimate PEP incidence—three complementary outputs that collectively inform both clinical application and model optimization.

## Clinical vpractice recommendations

**Model development.** Incorporate diverse predictors, account for potential interactions, and retain continuous variables where possible. We recommend building models using multicenter datasets that encompass a wide range of potential risk factors, to enhance representativeness and robustness.

**Advanced methods.** Integrate AI/ML techniques (e.g., random forest, support vector machines) alongside traditional logistic regression to capture complex relationships. Future research should explore whether combining traditional biostatistical approaches with AI/ML can yield better predictive performance than either method alone.

**Validation.** Conduct temporal, geographic, and domain-specific external validations using independent datasets. Among the included studies, 12 performed external validation, highlighting its feasibility.

**Implementation.** Present models via user-friendly tools (web calculators, mobile apps) and integrate with electronic health record systems to automate data entry and support real-time decision-making. Notably, one study transformed its model into an interactive online visualization platform for clinical use, which could serve as a practical example for promoting clinical adoption.

## Limitation

Our analysis is limited by the predominance of observational studies, variable reporting quality, and frequent absence of calibration or external validation in included models. Restricting to English and Chinese literature may have excluded relevant studies. Furthermore, high overall risk of bias in PROBAST assessments means that most models require further refinement and validation before routine clinical use.

## Conclusion

This study included 24 studies, encompassing 26 PEP risk prediction models. The results showed that most of the existing prediction models had good predictive performance, but the overall risk of bias was high. It is recommended that future research strictly follow the TROPID guidelines, conduct multicenter and large-sample studies to develop high-quality predictive model research to provide reference for clinical decision-making.

## Supporting information

**S1 Table. PRISMA 2020 checklist.**
(DOCX)

**S2 Table. PICOTS framework.**
(DOCX)

**S3 Table. Complete list of search terms.**
(DOCX)

**S4 Table. Risk of bias assessment using the Newcastle Ottawa Scale.**
(DOCX)

**S5 Table. GRADE assessment for post-endoscopic retrograde cholangiopancreatography pancreatitis.**
(DOCX)

**S1 Fig. Forest plot of Post-ERCP Pancreatitis incidence: a meta-analysis.**
(DOCX)

**S2 Fig. Forest plot of Post-ERCP Pancreatitis incidence: subgroup meta-analysis by study design.**
(DOCX)

**S3 Fig. Forest plot of Post-ERCP Pancreatitis incidence: subgroup meta-analysis by diagnostic criteria.**
(DOCX)

**S4 Fig. Forest plot of Post-ERCP Pancreatitis incidence: subgroup meta-analysis by geographic region.**
(DOCX)

**S5 Fig. Forest plot of Post-ERCP Pancreatitis incidence: subgroup meta-analysis by validation method.**
(DOCX)

**S6 Fig. Forest plot of the meta-analysis on risk factors for Post-ERCP Pancreatitis.**
(DOCX)

**S7 Fig. Forest plot of the meta-analysis on predictive performance for Post-ERCP Pancreatitis.**
(DOCX)

## Author contributions

**Conceptualization:** Yijun Mao, Cheng Zhang, Erqing Li.

**Data curation:** Yijun Mao.

**Formal analysis:** Yijun Mao.

**Funding acquisition:** Yijun Mao, Qiang Liu, Hui Fan, Wenjing He, Xueqian Ouyang, Erqing Li, Xiaojuan Wang, Li Qiu, Huanni Dong.

**Investigation:** Yijun Mao.

**Methodology:** Yijun Mao, Cheng Zhang.

**Project administration:** Yijun Mao, Li Qiu.

**Resources:** Yijun Mao.

**Software:** Yijun Mao.

**Supervision:** Yijun Mao, Qiang Liu, Hui Fan, Wenjing He, Cheng Zhang, Xueqian Ouyang, Xiaojuan Wang.

**Validation:** Yijun Mao, Huanni Dong.

**Visualization:** Yijun Mao.

**Writing – original draft:** Yijun Mao.

**Writing – review & editing:** Yijun Mao, Qiang Liu.

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
