## [Decision Letter · Decision Letter 0]

18 Jun 2025

Dear Dr. Liu,

Thank you for submitting your manuscript to PLOS ONE. After careful consideration, we feel that it has merit but does not fully meet PLOS ONE’s publication criteria as it currently stands. Therefore, we invite you to submit a revised version of the manuscript that addresses the points raised during the review process.

We look forward to receiving your revised manuscript.

Kind regards,

Suphakarn Techapongsatorn, M.D., Ph.D.

Academic Editor

PLOS ONE

Journal Requirements:

This research project was supported by Xianyang Science and Technology Planning Project (Grant Number: L2023-ZDYF-SF-055).

4. Please remove all personal information, ensure that the data shared are in accordance with participant consent, and re-upload a fully anonymized data set.

Reviewers' comments:

Reviewer's Responses to Questions

**Comments to the Author**

1. Is the manuscript technically sound, and do the data support the conclusions?

Reviewer #1: Partly

2. Has the statistical analysis been performed appropriately and rigorously?

Reviewer #1: Yes

3. Have the authors made all data underlying the findings in their manuscript fully available?

Reviewer #1: Yes

4. Is the manuscript presented in an intelligible fashion and written in standard English?

Reviewer #1: Yes

Reviewer #1: This meta-analysis paper has a clear overall research objective, relatively standardized research methods, and rich content. However, there are still some aspects that can be improved, which are as follows:

1. The inclusion criteria could be optimized. The included studies are only observational, which limits the diversity of the model sources. It is advisable to consider incorporating some interventional studies to comprehensively evaluate the performance of the model.

2. Insufficient consideration of multi - factor interactions: When exploring predictive factors, only the relationship between a single factor and PEP was analyzed, without considering the interactions among factors. It is recommended to supplement relevant analyses to more accurately reveal the disease risk mechanisms.

3. The search strategy can be refined. The search terms only involve common keywords, which may lead to the omission of relevant literature. It is recommended to supplement synonyms and abbreviations and search more databases, such as ClinicalTrials.gov and Google Scholar, to ensure comprehensiveness.

4. Limitations of the quality assessment tool: Only the PROBAST tool is used to assess the risk of bias. It would be beneficial to incorporate other tools, such as the Jadad quality scoring scale, to more accurately evaluate the quality of studies from multiple dimensions.

5. Analysis of model performance indicators is lacking: When reporting model performance, the analysis of the C - statistic and calibration results is rather superficial. It is necessary to delve deeper into the reasons for the performance differences among different models and the impact of these differences on clinical applications.

6. Currently, the paper has not conducted GRADE grading on the quality of evidence. It is recommended that, based on the GRADE system, the included studies be comprehensively evaluated in terms of research design, risk of bias, inconsistency, indirectness, publication bias, etc., and the quality grades of each study and the overall evidence be provided. Through GRADE grading, readers can more intuitively understand the reliability of the evidence, providing clearer guidance for subsequent clinical decision-making and further research.

7. Please supplement the detailed retrieval strategies for each database to enhance the scientific nature and reproducibility of the research.

8. Discussion Section --- Deficiencies in Comparison with Existing Studies: When discussing the model performance and predictive factors, there is a lack of sufficient comparison with similar studies, making it impossible to highlight the advantages and disadvantages of this study. It is recommended to supplement relevant comparative analysis to clarify the positioning of the study.

Clinical recommendations lack specific implementation paths: The proposed clinical practice recommendations are rather general, such as using AI and ML methods, conducting model validation, etc., and lack specific implementation steps and precautions. It is recommended to refine the recommendations to enhance their operability.

**Do you want your identity to be public for this peer review?** For information about this choice, including consent withdrawal, please see our Privacy Policy

Reviewer #1: No

---

## [Author Response · Author response to Decision Letter 1]

11 Jul 2025

Response to Journal Requirements

Dear Dr. Suphakarn Techapongsatorn,

Thank you for your constructive feedback. We have carefully addressed all the journal's additional requirements as follows:

1. Manuscript Formatting

We have reformatted the manuscript using the PLOS ONE style templates. All files have been renamed according to journal conventions.

2. ORCID iD Validation

The corresponding author's ORCID iD (0009-0006-1757-183X]) has been validated in Editorial Manager via the 'Fetch/Validate' function.

3. Funder's Role Statement (Amended in manuscript)

The updated declaration reads: "This work was supported by Xianyang Science and Technology Planning Project (L2023-ZDYF-SF-055). The funders had no role in study design, data collection and analysis, decision to publish, or preparation of the manuscript." �Page 25, Line 530-531

4. Data Anonymization

The dataset used in this study is completely anonymized and contains no personal identifying information.

5. Supporting Information

Added captions at the manuscript end (before References):

S1 Table. PRISMA 2020 Checklist.

S2 Table. PICOTS Framework.

S3 Table. Complete List of Search Terms.

S4 Table. Risk of bias assessment using the Newcastle Ottawa Scale.

S5 Table. GRADE assessment for post-endoscopic retrograde cholangiopancreatography pancreatitis.

All changes have been verified by all co-authors. Thank you for considering our revised submission.

Sincerely,

Yijun Mao

Xianyang Central Hospital

hlb33288602@163.com

Dear Dr. Suphakarn Techapongsatorn,

Thank you for giving us the opportunity to revise our manuscript [ Manuscript ID: PONE-D-24-50343, Title: Risk prediction model for post-endoscopic retrograde cholangiopancreatography pancreatitis: A systematic review and meta-analysis ]. We sincerely appreciate the reviewers’ constructive comments, which have helped us significantly improve the paper.

We have carefully addressed all comments point-by-point in this response letter. Major revisions are highlighted in yellow in the manuscript.

We believe the revised manuscript now meets the journal’s standards. Please find our detailed responses below.

Sincerely,

Yijun Mao, Qiang Liu, Hui Fan, Wenjing He, Cheng Zhang, Xueqian Ouyang, Erqing Li, Xiaojuan Wang, Li Qiu, Huanni Dong

Xianyang Central Hospital

hlb33288602@163.com

445829791@qq.com

Reviewer #1:

We thank the reviewer for their insightful suggestions.

Comment 1: The inclusion criteria could be optimized. The included studies are only observational, which limits the diversity of the model sources. It is advisable to consider incorporating some interventional studies to comprehensively evaluate the performance of the model.

Response:

We sincerely appreciate this constructive suggestion. In response:

1. Criteria Modification:

We have revised the inclusion criteria to explicitly allow for interventional studies containing PEP prediction models (Section Inclusion and exclusion criteria, Page 6). This change enhances the scope while maintaining our focus on model validation.

2. Current Study Rationale:

Our initial restriction to observational studies aimed to ensure homogeneity in model derivation populations, as most ERCP prediction models are developed using observational data.[1-3]

3. Future Direction:

We fully agree that interventional studies could provide valuable insights into model performance under controlled conditions. This limitation is now acknowledged in the Discussion (Page 24, Lines 496-499).

Changes made:

- Inclusion criteria updated in Methods

- Limitations section expanded

- Added references supporting observational data predominance in model development

Comment 2: Insufficient consideration of multi - factor interactions: When exploring predictive factors, only the relationship between a single factor and PEP was analyzed, without considering the interactions among factors. It is recommended to supplement relevant analyses to more accurately reveal the disease risk mechanisms.

Response:

We sincerely appreciate this insightful suggestion. We have now supplemented the analysis of multifactor interactions as follows:

1. Mechanistic discussion:

Added a dedicated subsection (Page 20-21) explaining how factor interactions may synergistically increase PEP risk through:

• Mechanical trauma compounding (e.g., repeated cannulation + guidewire)

• Physiological susceptibility modulation (e.g., gender-specific sphincter reactivity)

• Metabolic stress amplification (e.g., hypoalbuminemia + prolonged ischemia)

Comment 3: The search strategy can be refined. The search terms only involve common keywords, which may lead to the omission of relevant literature. It is recommended to supplement synonyms and abbreviations and search more databases, such as ClinicalTrials.gov and Google Scholar, to ensure comprehensiveness.

Response:

We sincerely appreciate the reviewer’s constructive suggestions to improve the literature search comprehensiveness.

1. Database expansion:

Added ClinicalTrials.gov and Google Scholar per the reviewer’s suggestion.

2. Validation:

Revised search strategy yielded no additional eligible studies.

3.Supplemented the detailed retrieval strategies for all databases in:

S2 Table. PICOTS Framework

S3 Table. Complete List of Search Terms

4.Added the following elements to ensure reproducibility:

Complete search syntax with Boolean operators

Both controlled vocabulary (e.g., MeSH) and free-text terms

Number of records retrieved from each database

5.Cited these supplementary tables in the Methods section (subsection "Search strategy", Page 5, Line 107).

Changes made:

S2 Table and S3Table: Provided complete search syntax for all databases.

We confirm these modifications have significantly enhanced the study’s methodological rigor.

Comment 4: Limitations of the quality assessment tool: Only the PROBAST tool is used to assess the risk of bias. It would be beneficial to incorporate other tools, such as the Jadad quality scoring scale, to more accurately evaluate the quality of studies from multiple dimensions.

Response:

We sincerely appreciate this constructive suggestion. To address the concern about multidimensional quality evaluation, we have implemented the following improvements:

1.Methodological Enhancement

Added Newcastle-Ottawa Scale (NOS) for observational studies alongside PROBAST

This provides complementary perspectives:

✓ PROBAST: Model-specific risk of bias

✓ NOS: General study quality metrics

2.Results Expansion

Created S4 Table comparing quality ratings across studies (Page 13, Lines 274-277)

3.Transparent Reporting

Clearly labeled the secondary role of NOS in manuscript (Page 7, Lines 141-143)

Changes made:

Methods: Page 7, Section Quality assessment

Results: Page 13, Lines 274-277, New S4 Table

We believe these modifications have strengthened the methodological rigor while maintaining focus on prediction model-specific assessment.

Comment 5: Analysis of model performance indicators is lacking: When reporting model performance, the analysis of the C - statistic and calibration results is rather superficial. It is necessary to delve deeper into the reasons for the performance differences among different models and the impact of these differences on clinical applications.

Response:

We sincerely appreciate this constructive suggestion. We have significantly expanded the analysis of model performance in the revised manuscript:

1.Stratified analysis of C-statistic performance with high/low performance model comparisons.

2.Detailed examination of predictor categories contributing to discrimination differences

3.Calibration assessment with both statistical tests and graphical analysis

4.Clinical applicability discussion based on model complexity

Changes made:

Revised text in Section Model performance (Page 10-11).

Comment 6: Currently, the paper has not conducted GRADE grading on the quality of evidence. It is recommended that, based on the GRADE system, the included studies be comprehensively evaluated in terms of research design, risk of bias, inconsistency, indirectness, publication bias, etc., and the quality grades of each study and the overall evidence be provided. Through GRADE grading, readers can more intuitively understand the reliability of the evidence, providing clearer guidance for subsequent clinical decision-making and further research.

Response:

We sincerely appreciate this constructive suggestion. Following the GRADE methodology, we have:

1.Added a dedicated evaluation section in Methods (Page 7, Lines 143-146)

2.Created S5 Table summarizing quality assessment for all outcomes

3.The GRADE classification outcomes and a detailed analysis of each dimension in Results (Page 13-14)

Key Findings:

The overall evidence quality was rated as moderate due to:

Partial studies failed to report model calibration details, resulting in an overall moderate risk of bias (downgraded for risk of bias)

Substantial heterogeneity in model discrimination was observed (I²=84%, downgraded for inconsistency)

Changes made:

Methods: Page 7, Lines 143-146

Results: Page 13-14, New S5 Table

Comment 7: Please supplement the detailed retrieval strategies for each database to enhance the scientific nature and reproducibility of the research.

Response:

1.Supplemented the detailed retrieval strategies for all databases in:

S2 Table. PICOTS Framework

S3 Table. Complete List of Search Terms

2.Added the following elements to ensure reproducibility:

Complete search syntax with Boolean operators

Both controlled vocabulary (e.g., MeSH) and free-text terms

Number of records retrieved from each database

3.Cited these supplementary tables in the Methods section (subsection "Search strategy", Page 5, Line 107).

Changes made:

S2 Table and S3 Table: Provided complete search syntax for all databases.

Comment 8: Discussion Section --- Deficiencies in Comparison with Existing Studies: When discussing the model performance and predictive factors, there is a lack of sufficient comparison with similar studies, making it impossible to highlight the advantages and disadvantages of this study. It is recommended to supplement relevant comparative analysis to clarify the positioning of the study.

Response:

We have added a dedicated subsection (Page 21-22) comparing our findings with 4 key previous reviews[4-7]. The new analysis:

1)Quantifies performance evolution (AUC improvement trends)

2)Identifies paradigm shifts in predictor selection

Changes made:

Added comparative analysis: Page 21-22.

New citations supporting the comparison: References 59-62.

Comment 9: Clinical recommendations lack specific implementation paths: The proposed clinical practice recommendations are rather general, such as using AI and ML methods, conducting model validation, etc., and lack specific implementation steps and precautions. It is recommended to refine the recommendations to enhance their operability.

Response:

To refine clinical implementation recommendations, we have further specified: (1) the procedural steps for applying AI and machine learning methodologies, (2) concrete approaches for conducting external model validation, and (3) detailed strategies for clinical deployment through web-based calculators or applications, thereby ensuring operational feasibility and generalizability.

Changes made:

Replaced generic suggestions with actionable protocols: Page 22-23.

Declaration: All co-authors have approved the revised manuscript and our responses to the reviewers.

References:

[1]. Collins, G.S., et al., Transparent reporting of a multivariable prediction model for individual prognosis or diagnosis (TRIPOD): the TRIPOD statement. BMJ, 2015. 350: p. g7594.

[2]. Dreyer, N.A., et al., Why observational studies should be among the tools used in comparative effectiveness research. Health Aff (Millwood), 2010. 29(10): p. 1818-25.

[3]. Steyerberg, E.W., et al., Prognosis Research Strategy (PROGRESS) 3: prognostic model research. PLoS Med, 2013. 10(2): p. e1001381.

[4]. Beran, A., et al., Predictors of Post-endoscopic Retrograde Cholangiopancreatography Pancreatitis: A Comprehensive Systematic Review and Meta-analysis. Clinical Gastroenterology and Hepatology.

[5]. Sperna Weiland, C.J., et al., Preventive Measures and Risk Factors for Post-ERCP Pancreatitis: A Systematic Review and Individual Patient Data Meta-Analysis. Digestive Diseases and Sciences, 2024. 69(12): p. 4476-4488.

[6]. Sabrie, N., et al., Performance of Clinical Risk Prediction Models for Post-ERCP Pancreatitis: A Systematic Review. Pancreas, 9900.

[7]. Ding, X., F. Zhang and Y. Wang, Risk factors for post-ERCP pancreatitis: A systematic review and meta-analysis. Surgeon, 2015. 13(4): p. 218-29.

---

## [Decision Letter · Decision Letter 1]

9 Aug 2025

Dear Dr. Liu,

Thank you for submitting your manuscript to PLOS ONE. After careful consideration, we feel that it has merit but does not fully meet PLOS ONE’s publication criteria as it currently stands. Therefore, we invite you to submit a revised version of the manuscript that addresses the points raised during the review process.

We look forward to receiving your revised manuscript.

Kind regards,

Suphakarn Techapongsatorn, M.D., Ph.D.

Academic Editor

PLOS ONE

Journal Requirements:

Reviewers' comments:

Reviewer's Responses to Questions

**Comments to the Author**

Reviewer #1: All comments have been addressed

Reviewer #2: All comments have been addressed

2. Is the manuscript technically sound, and do the data support the conclusions?

Reviewer #1: Yes

Reviewer #2: No

3. Has the statistical analysis been performed appropriately and rigorously?

Reviewer #1: Yes

Reviewer #2: No

4. Have the authors made all data underlying the findings in their manuscript fully available?

Reviewer #1: Yes

Reviewer #2: Yes

5. Is the manuscript presented in an intelligible fashion and written in standard English?

Reviewer #1: Yes

Reviewer #2: Yes

Reviewer #1: The article has been revised very well. It has certain scientific significance, so I suggest it be published.

Reviewer #2: This paper aims to systematically review the methodological quality and predictive performance of risk prediction models for post-endoscopic retrograde cholangiopancreatography pancreatitis (PEP), analyze the predictors used in these models, and assess the risk of bias and clinical applicability. Overall, the structure is chaotic, the content of each part is not detailed enough. More importantly, the author needs to deeply reflect and clarify whether the purpose of this study matches the method of systematic review. From the review of the current research situation, the author emphasized the importance of predictive models and proposed that there are differences in the quality of different models. Therefore, a systematic review is necessary. In fact, the factors included in predictive models vary in different studies, and the existence of differences is certain. Therefore, from the perspective that the systematic review aims to integrate the results of various studies, the author needs to further clarify the matching between the purpose and the methods when discussing the significance of the research. Personally, I am not very enthusiastic about the meta-analysis of predictive models. However, based on the existing methodological framework, the authors need to incorporate the following three core results: ① Meta-analysis of the efficacy of predictive models; ② Meta-analysis of the incidence of post-endoscopic retrograde cholangiopancreatography pancreatitis (PEP) ③ Meta-analysis results of each influencing factor.

**Do you want your identity to be public for this peer review?** For information about this choice, including consent withdrawal, please see our Privacy Policy

Reviewer #1: No

Reviewer #2: No

---

## [Author Response · Author response to Decision Letter 2]

19 Aug 2025

Dear Dr. Suphakarn Techapongsatorn,

Thank you for your constructive feedback on our manuscript titled "Risk prediction model for post-endoscopic retrograde cholangiopancreatography pancreatitis: A systematic review and meta-analysis" (PONE-D-24-50343R1). We sincerely appreciate the time and effort invested by the academic editor and reviewers in evaluating our work.

We have carefully addressed all the comments and suggestions raised during the review process. Major revisions are highlighted in yellow in the manuscript. Below is a summary of the key revisions made:

Reviewer #1

The article has been revised very well. It has certain scientific significance, so I suggest it be published.

Response:

Thank you for your positive feedback on our revised manuscript. We greatly appreciate Reviewer 1’s acknowledgment of our revisions and the recommendation for publication. As suggested, we have thoroughly addressed all previous comments in the first round of revision, and we are delighted that the revised version meets the journal’s standards.

Reviewer #2

We sincerely thank the reviewer for the constructive and detailed feedback, which has significantly improved the clarity, methodological rigor, and clinical relevance of our manuscript. Below we provide a point-by-point response, with all changes incorporated into the revised version.

1. Concern about the match between study purpose and the systematic review method

Reviewer’s comment:

The author needs to deeply reflect and clarify whether the purpose of this study matches the method of systematic review. From the review of the current research situation, the author emphasized the importance of predictive models and proposed that there are differences in the quality of different models. Therefore, a systematic review is necessary. In fact, the factors included in predictive models vary in different studies, and the existence of differences is certain. Therefore, from the perspective that the systematic review aims to integrate the results of various studies, the author needs to further clarify the matching between the purpose and the methods when discussing the significance of the research.

Response:

We appreciate the reviewer’s observation. In the revised Introduction (page 5-6, lines 100–126) and Discussion (page 27, lines 573–579), we have explicitly clarified the methodological appropriateness of using a systematic review and meta-analysis for this topic. While heterogeneity in predictors is expected, such variability does not diminish the value of synthesis; rather, pooling allows us to (i) identify predictors with stable and clinically meaningful effects, (ii) quantify the predictive performance of models, and (iii) estimate the pooled incidence of PEP. This integrated approach provides a robust evidence base for refining models and guiding clinical application, thereby aligning the study purpose with the chosen methodology.

2. Need for three core quantitative results

Reviewer’s comment:

Based on the existing methodological framework, the authors need to incorporate the following three core results: ① Meta-analysis of the efficacy of predictive models; ② Meta-analysis of the incidence of PEP; ③ Meta-analysis results of each influencing factor.

Response:

We have fully incorporated these three core analyses into the Methods, Results, and associated figures/tables:

1.Meta-analysis of the incidence of PEP — We pooled data from 37 studies (n=38,016 patients) using a random-effects model with logit transformation, providing overall and subgroup estimates (page 12-13, lines 245–277; Figure 3, Figure S1-5).

2.Meta-analysis of individual predictors — For predictors reported in ≥3 studies, we extracted effect sizes (OR/RR/HR) and pooled them using random-effects models, identifying those with consistent, statistically significant associations (page 13-14, lines 280–304; Table 3, Figure S6A-K).

3.Meta-analysis of predictive model performance — We synthesized AUCs using random-effects and bivariate models, with results presented in forest plots (page 14-17, lines 306–360; Figure 4, Figure S7A-H).

Statistical methods for each analysis are detailed in the Statistical Analysis section (page 9–10, lines 181-208), including heterogeneity assessment, subgroup/sensitivity analyses, and publication bias evaluation.

3. Structural clarity and content detail

Reviewer’s comment:

Overall, the structure is chaotic, the content of each part is not detailed enough.

Response:

We have substantially restructured the manuscript to improve clarity and coherence:

Introduction — Now explicitly outlines the clinical background, research gap, necessity of a systematic review, and alignment of aims with methods (page 4–6, lines 88-126).

Results — Divided into three clearly labeled subsections corresponding to the three meta-analyses, each with summary statistics, heterogeneity measures, and visualizations (page 12–17, lines 245-360).

Discussion — Reorganized into three principal findings, each aligned with a results subsection, followed by methodological implications, strengths/limitations, and clinical/research recommendations(page 22–28, lines 470-603).

All figures, tables, and supplementary materials are now cross-referenced in the text to improve navigability.

4. Additional methodological rigor

We followed PRISMA 2020 and PROBAST guidelines, and explicitly described data extraction, risk-of-bias assessment, and meta-analytic methods (including transformations, model choice, and effect size conversions). This ensures reproducibility and transparency in accordance with best practices for prediction model reviews.

Additional Revisions:

Updated the PRISMA checklist (Table S1).

Added a supplemental file with expanded data on incidence, risk factors, and predictive performance.

We believe these revisions have significantly improved the manuscript and hope it now meets PLOS ONE’s publication standards. The revised files include:

Response to Reviewers (this letter).

Revised Manuscript with Track Changes (highlighting all edits).

Manuscript (clean version).

Should further modifications be required, we are happy to comply. Thank you again for your consideration.

Sincerely,

Yijun Mao, Qiang Liu, Hui Fan, Wenjing He, Cheng Zhang, Xueqian Ouyang, Erqing Li, Xiaojuan Wang, Li Qiu, Huanni Dong

Xianyang Central Hospital

hlb33288602@163.com

---

## [Decision Letter · Decision Letter 2]

31 Aug 2025

Risk prediction model for post-endoscopic retrograde cholangiopancreatography pancreatitis: A systematic review and meta-analysis

PONE-D-24-50343R2

Dear Dr. Qiang Liu, 

We’re pleased to inform you that your manuscript has been judged scientifically suitable for publication and will be formally accepted for publication once it meets all outstanding technical requirements.

Kind regards,

Suphakarn Techapongsatorn, M.D., Ph.D.

Academic Editor

PLOS ONE

Additional Editor Comments (optional):

Reviewers' comments:

Reviewer's Responses to Questions

**Comments to the Author**

Reviewer #3: (No Response)

2. Is the manuscript technically sound, and do the data support the conclusions?

Reviewer #3: Yes

3. Has the statistical analysis been performed appropriately and rigorously?

Reviewer #3: Yes

4. Have the authors made all data underlying the findings in their manuscript fully available?

Reviewer #3: Yes

5. Is the manuscript presented in an intelligible fashion and written in standard English?

Reviewer #3: Yes

Reviewer #3: The details outlined in the manuscript are relevant, and all weaknesses of the concluded results have been highlighted for the reader to consider and remain cautious about when interpreting the study’s findings. This systematic review has scientific significance in guiding clinical decision-making and informing future model development for PEP. I believe this manuscript has enough scientific merit to be published.

**Do you want your identity to be public for this peer review?** For information about this choice, including consent withdrawal, please see our Privacy Policy

Reviewer #3: No

---

## [Editor Report · Acceptance letter]

PONE-D-24-50343R2

PLOS ONE

Dear Dr. Liu,

I'm pleased to inform you that your manuscript has been deemed suitable for publication in PLOS ONE. Congratulations! Your manuscript is now being handed over to our production team.

Kind regards,

on behalf of

Dr. Suphakarn Techapongsatorn

Academic Editor

PLOS ONE